# Poly(vinylidene fluoride) Intestinal Sleeve Implants for the Treatment of Obesity and Type 2 Diabetes

**DOI:** 10.3390/polym14112178

**Published:** 2022-05-27

**Authors:** Hao-Ming Chang, Wei-Ping Zhan, Hsieh-Chih Tsai, Meng-Ru Yang

**Affiliations:** 1Division of General Surgery, Tri-Service General Hospital, National Defense Medical Center, Taipei 114, Taiwan; doc20364@gmail.com; 2Graduate Institute of Applied Science and Technology, National Taiwan University of Science and Technology, Taipei 106, Taiwan; ni47ke@gmail.com (W.-P.Z.); ernie.mr.yang@gmail.com (M.-R.Y.); 3Advanced Membrane Materials Center, National Taiwan University of Science and Technology, Taipei 106, Taiwan; 4R&D Center for Membrane Technology, Chung Yuan Christian University, Chungli, Taoyuan 320, Taiwan

**Keywords:** poly(vinylidene fluoride) (PVDF), piezoelectric material, solution casting, machine direction orientation (MDO), intestinal sleeve implant, type 2 diabetes

## Abstract

Currently, treatment of diabetes and associated obesity involves Roux-en-Y gastric bypass or sleeve gastrectomy to reduce the absorption of nutrients from the intestine to achieve blood glucose control. However, the surgical procedure and subsequent recovery are physically and psychologically burdensome for patients, with possible side effects, so alternative treatments are being developed. In this study, two methods, solution casting and machine direction orientation (MDO), were used to prepare intestinal implants made of poly(vinylidene fluoride) (PVDF) film and implant them into the duodenum of type 2 diabetic rats for the treatment of obesity and blood glucose control. The PVDF film obtained by the MDO process was characterized by FT-IR, Raman spectroscopy, XRD and piezoelectricity tests, which showed higher composition of β crystalline phase and better elongation and mechanical strength in specific directions. Therefore, the material was finally tested on rats after it was proven to be non-toxic by biological toxicity tests. The PVDF was implanted into alloxan-induced diabetic rats, which were used as a model of impaired insulin secretion due to pancreatic beta cell destruction rather than obesity-induced diabetes, and rats were tracked for 24 days, showing significantly improved body weight and blood glucose levels. As an alternative therapeutic option, intestinal sleeve implant showed future potential for application.

## 1. Introduction

Due to changes in modern eating habits and lifestyles, as well as population growth and aging, the global prevalence of diabetes has doubled from 4.7% in 1980 to 8.5% in 2014. The World Health Organization (WHO) states that in 2019, diabetes was considered the cause of death for 1.5 million people. Type 2 diabetes accounted for nearly 90% of cases, which is physically and mentally painful for patients and also places a significant burden on the health care system [1,2,3]. The obesity problem that often accompanies type 2 diabetes is predicted to cost the U.S. $860 billion in health care spending by 2030 [4], with even more serious consequences if patients are not treated [5,6,7,8,9]. Roux-en-Y gastric bypass and sleeve gastrectomy are the most common surgical treatments. The former is widely used in patients with type 2 diabetes and severe esophageal reflux, but there are risks such as malnutrition, intestinal obstruction, and inability to perform endoscopy through the mouth, as well as various surgical risks. Sleeve gastrectomy has increased in recent years because of its relatively low surgical complexity, with advantages that include a lower risk of anemia and no need for continuous additional nutrient supplementation, but there are still problems such as regaining weight due to the enlarged gastric pouch and risks associated with surgery [10,11,12,13,14,15,16]. Some patients still have concerns about the procedure given the possible lifelong side effects or irreversibility, which has prompted more attention to and development of the surgical implant technique.

In the past, the main surgical implants were a BioEnteriscs intragastric balloon (BIB) and laparoscopic adjustable gastric banding (LAP Band). In the former, patients were usually anesthetized and a silicone ball was placed in the stomach through the esophagus with the assistance of a gastroscope, without additional surgery. It was a relatively safe method, but it can only be used for short-term transition, and it was mainly expected that the patients themselves would develop good eating and living habits to achieve long-term improvement [17,18,19]. In the latter case, a silicone band was used to separate a small space in the upper part of the stomach, and the tightness of this band was adjusted by a regulator buried under the belly, so that the small space would be filled with adequate amounts of food to create a feeling of satiety. However, it would take approximately one year to adjust the band slowly after surgery, and there was a possibility of post-operative infection or dislocation that would require a second surgery [20,21,22]. Intestinal sleeve implant was a relatively new technique, and the main commercialized product was EndoBarrier^®^ from GI Dynamics. It was composed of a titanium alloy fixture at the upper end, combined with a 60 cm-long fluoropolymer film sleeve at the lower end, which can be implanted through an endoscope and formed a physical barrier in the intestine to reduce the direct absorption of food nutrients and alter the secretion of human hormones (Figure 1) [23]. When applied to patients with obesity and type 2 diabetes, it provided effective improvements in blood glucose and weight over a few months to a year of trials. Rapid delivery of nutrients to the distal gut can exclude nutrients from the proximal bowel as well as duodenal exclusion and gut hormonal changes, resulting in elevated glucagon-like peptide-1 (GLP-1) and peptide YY (PYY) and improvements in insulin resistance, glucose tolerance, and beta-cell function, likely contributing to glucose homeostasis [24,25]. Therefore, more studies on material selection and design optimization were motivated by the good therapeutic performance of this alternative treatment method.

Currently, many biomedical materials have been developed for human implants with different properties such as biodegradability, bioactivity and bioinertness [26]. However, utilization of piezoelectric materials in biomedical device fabrications is rarely handled because of its practical limitations such as biological compatibility, flexibility and structure of the biological systems. Polymer-based piezoelectric materials have more advantages in fabricating biomedical devices compared with inorganic-based piezoelectric materials such as barium titanate (BaTiO_3_), quartz, aluminum nitrate (AIN), and zirconium titrates (PZT) [27,28,29]. Particularly, poly(vinylidene fluoride) (PVDF) is a semi-crystalline piezoelectric polymer with a glass transition temperature (T_g_) of −35 °C and a melting point (T_m_) of 177 °C, and provides better stability and resistance to acid and alkali corrosion. PVDF can be divided into five crystalline phases—α, β, γ, δ and ε—and is converted by heat compression [29,30], additional electric field [31,32], quenching [33,34], and other processes. Some of the phases even have different piezoelectric properties due to changes in polarity, and so they have been widely used in the fields of sensors, energy and biomedicine. In this study, two different processes, solution casting and machine direction orientation (MDO), were used to produce film intestinal sleeves with different PVDF crystalline phase ratios, which were implanted into rats with type 2 diabetes to evaluate the effect on weight, blood glucose control and therapeutic efficacy. These intestinal sleeve implants may have high potential for treatment of obesity and type 2 diabetes in clinical practice.

The aim of this work is to prepare a PVDF sleeve using two methods: solution casting and machine direction orientation. Newly prepared films were thermally treated and characterized by FTIR, RAMAN, SAXS, SEM, mechanical testing and ESI. Thermal treatment induced changes in the crystalline structure of the polymer, mainly resulting in a beta-crystalline structure due to transformation of the alpha-crystalline structure. Finally, experiments were performed using SD rats, resulting in a weight reduction of 11% after 24 days, which is very similar to the commercially available EndoBarrier.

## 2. Experimental Sections

### 2.1. Materials

Poly(vinylidene fluoride) (PVDF), N-methylpyrrolidone (NMP), and alloxan monohydrate were all purchased from Sigma-Aldrich in Saint Louis, MO 63103, USA. All other chemicals purchased were of analytical grade and without further purification.

### 2.2. Preparation of PVDF Films

PVDF films were prepared by two different methods. The first method was solution casting. An amount of 10 g of PVDF was dissolved in 40 g of NMP and stirred at 500 rpm at 100 °C for at least 6 h until the solution was visually clear. Then, the solution was poured onto a smooth glass substrate and quickly squeegeed with a coater (Appendix A), and then dried in an oven at 90 or 100 °C for 24 h to remove the solvent. After drying, the glass substrate was cooled at room temperature for 30 min before tearing off to avoid deformation of the film by pulling it at a high temperature.

The second method was MDO. First, the PVDF powder was heated in a single-screw extruder at 100 °C to process the powder into wire. Second, the obtained PVDF wire was directly placed into the stretching machine and preheated in rolls at 90 or 100 °C, then in rolls with a stretching ratio of 3 by adjusting the speed for machine direction orientation, and finally annealed in rolls at 25 °C and cooled and shaped to obtain a PVDF film.

### 2.3. Characterization of PVDF Films

The surface morphology of the films was analyzed by scanning electron microscopy (SEM) (JEOL, JSM-6500F, JEOL LTD. Tokyo, Japan) at 10 kV. The crystalline phases were analyzed by Fourier transform infrared (FTIR) spectroscopy (PerkinElmer Spectrum two, PerkinElmer Ltd, Bucks, UK) and Raman spectroscopy (JASCO NRS5100) with a laser light wavelength of 532 nm. In addition, a universal tensile tester (Shimadzu EZ-LX, Shimadzu® Corp, Kyoto, Japan) was used to measure the mechanical properties of the films. All of the samples were molded into dumbbell shapes with a total length of 165 mm and thickness of 0.4 mm (two shoulder sections were 54 mm in length and 19 mm in width; the middle section was 57 mm in length and 13 mm in width), and tested with a strain rate of 10 mm/min. The stress was obtained by dividing the force by the cross-section area, the strain as the length change related to the original length, and Young’s modulus as the slope of the stress–strain curve in the linear zone. The piezoelectric properties were measured by electrochemical impedance spectroscopy (EIS) combined with an oscilloscope.

### 2.4. Preparation of PVDF Film Intestinal Sleeve Implants

To prepare the intestinal sleeve implant, it was necessary to combine two components (Figure 2, left side). One part was a 1 cm diameter thermoplastic polyurethane (TPU) flat-bottomed spherical hollow model that serves as a fixation device for the anterior end of the duodenum of the rat after implantation. In the other part, the film was rolled into a cylinder of 1 cm diameter and 5 cm length and sealed with a heat sealer at the joints. Finally, the two parts were bundled and connected with non-absorbable sutures to obtain the PVDF film intestinal sleeve implant.

### 2.5. Animal Experiments

The Sprague Dawley rats (SD rats) used in the experiments were obtained from the National Defense Medical Center Animal Center. The animal experiments have been reviewed by the Institutional Animal Care and Use Committee and follow the guide for the care and use of laboratory animals. The rats were first induced with alloxan monohydrate at a dose of 170 mg/kg. The SD rats were then monitored for fasting glucose >120 mg/dL for 3 days to ensure successful induction of type 2 diabetes before surgery. Animal experiments were divided into five groups: solution casting at 90 °C, solution casting at 100 °C, MDO at 90 °C, MDO at 100 °C, and control. Each group had 5 rats, hence the total number of rats involved in animal study experiment was 25.

To compare film intestinal sleeve implants obtained with different preparation methods, the trial was divided into three groups. One group was SD rats implanted with film intestinal sleeve implants prepared by the solution casting method, and the other group was SD rats implanted with film intestinal sleeve implants prepared by the MDO method. The remaining group of SD rats were not implanted with film intestinal sleeve implants but underwent the same surgical procedure as the control group. The specific surgical procedure was to ensure that the SD rats were fasted for at least 18 h, followed by intravenous injection of Zoletil 50 at a dose of 10 mg/kg to complete general anesthesia and fixation on the surgical table. After opening the abdominal cavity by midline dissection, a 10 mm incision was made in the lower part of the stomach near the pylorus, and then the film intestinal sleeve implant was slowly inserted into the duodenum through the pylorus using an auxiliary device until the fixation device was approximately 5 mm below the pylorus. The wound was then sutured and completed (Figure 2a–d). Blood glucose and body weight of SD rats were measured at 0, 12 and 24 days after completion of surgery.

### 2.6. The Oral Glucose Tolerance Test

The oral glucose tolerance test (OGTT) is used to measure the ability of an organism to absorb glucose in order to measure insulin function and glucose tolerance. Type 2 diabetes SD rats were fasted for 18 h to ensure that they were in a fasting state. Blood was first collected from the tail vein to measure the 0 min glucose value, and then 2 mg/mL of glucose was injected orally into the rats. After that, blood was collected at 30, 60, 90 and 120 min and blood glucose levels were measured again.

## 3. Result and Discussion

### 3.1. Characterization of PVDF Films

The PVDF films prepared by the solution casting method at 90 and 100 °C were observed at 500× and 5000× using a scanning electron microscope. First, at a magnification of 500×, the surface of the film had a certain degree of unevenness and porosity at both temperatures (Figure 3a,b). However, when combined with the images at 5000×, further analysis revealed that the number and size of pores tend to decrease as the temperature is increased, and show a flatter and denser surface (Figure 3c,d). The crystalline structure of granular protrusions can be observed on the surface because the drying temperature of the PVDF solution coated on the glass plate in the oven was different and therefore would affect the volatilization rate of the solvent. At a lower temperature, due to the slower volatilization rate and longer residence time of the solvent, the film forms a spatial barrier during the drying process of the film, resulting in the formation of pore structures. The slower drying process also meant slower film formation, which leads to a certain degree of aggregation and inhomogeneity of the polymer solution, and eventually affects the structural compactness and flatness of the film. 

The PVDF films obtained after stretching at 90 and 100 °C using the MDO method were observed at 1000× and 5000× using a scanning electron microscope (Figure 3e–h). No obvious holes or particle-like protrusions were observed on the surface of the films, which were smoother than those from the solution casting method. The main reason for this was that during the MDO process, continuous and steady stress was applied to the surface of the material by stretching rollers, and therefore the process produced corresponding stress marks (Figure 3e,f). Furthermore, because of the unique properties of PVDF, its crystalline state can change at high temperatures and pressures, which also leads to changes in the microstructure.

In order to confirm the crystalline phase transition of PVDF more precisely, FTIR analysis was performed. For the films prepared by the solution casting method, the characteristic peaks of the γ phase were observed at 810 and 1234 cm^−1^ in the infrared spectrum. This was mainly due to the transformation of the α phase into the γ phase during the high temperature annealing process of the sample preparation. However, only part of the crystalline phase was converted, so the characteristic peaks of the α-phase were still observed at 1209 and 1180 cm^−1^.

For the films prepared by the MDO method, a high percentage of the α phase was converted to the β phase during the stretching process, so the characteristic peaks of the β phase can be observed at 839, 1275 and 1431 cm^−1^, but some weak characteristic peaks of the α phase can still be observed (Figure 4a). The wavenumber and vibration mode of the characteristic peaks of each phase are also listed for reference (Appendix A) [35,36,37,38,39].

After qualitative analysis by FTIR, we were able to confirm the presence of different crystalline phases, but there was no quantitative response to the proportion of each phase. Therefore, Raman spectroscopy was further used to analyze the phases and to perform peak fitting and peak separation. The characteristic peaks of each phase in the Raman spectra of the solution casting and MDO films have been labeled (Figure 4b and Appendix A) [40]. The peaks in the range of 780–860 cm^−1^ had a higher intensity and overlap, so they were selected for peak fitting and peak separation (Appendix A). Then, the peaks at 795, 840 and 812 cm^−1^ were selected to correspond to the α, β and γ phases, respectively, for the calculation of the crystalline phase ratio. The peak intensity of the β phase was divided by the sum of the intensity values of the α, β and γ phases to compare the results of the β crystalline phase ratio of the films obtained by different methods and parameters (Table 1). Similarly, the XRD results also supported the previous analysis. In the solution casting group, a strong peak at 19.9°, and weak peaks at 18.5° and 39.1° were observed, indicating higher γ phase composition. The MDO group, on the other hand, showed a strong peak at 20.2° and a weak peak at 36.3°, indicating higher β-phase composition (Figure 5a,b) [41]. It was found that the MDO group had a higher percentage of β crystalline phase, making the α phase convert to the β phase more efficiently, and also bringing more significant piezoelectric properties (Figure 5c–e). Presented in Figure 5c–e are the voltage pulses for both the solution casting and MDO approaches varying time and with an applied pressure of 1 N/cm^−1^. A significant voltage difference of approximately 0.4 V was obtained for MDO, which is attributed to the higher β phase. Similarly, with an applied pressure of 1 N/cm^−1^, a voltage difference of approximately 2 V can be generated.

Combining the above analyses of crystal phase composition, possible differences in the mechanical properties should be noted. The yield point of the solution casting method group reached 66.08 MPa and Young’s modulus reached 38.81 MPa as the process temperature was increased. This trend corresponds to the SEM image of the denser structure due to the increase in process temperature, and indicates better mechanical properties. The MDO method group, which was subjected to unidirectional stretching during the process, exhibited directional selectivity. In parallel stretching, the yield point reached 173.32 Mpa, but the average elongation was only 14.59%. In vertical stretching, the yield point was only 22.78 MPa, but the average elongation was up to 94.89% (Figure 6 and Table 2). Therefore, compared to the typical stress–strain curve, which exhibited a general positive correlation between stress and strain, structural changes generated by the MDO manufacturing process resulted in a separate increase in stress or strain in two separate tensile directions [42].

### 3.2. In Vivo PVDF Film Intestinal Sleeve Implantation for Glucose and Obesity Control 

Before in vivo testing, the material was analyzed using the 3-(4,5-dimethylthiazol-2-yl)-2,5-diphenyl-2H-tetrazolium bromide (MTT) assay and was shown to be non-cytotoxic (Appendix A). In order to compare the difference in blood glucose levels after implantation of the intestinal sleeve, an oral glucose tolerance test was used and the area under the curve was calculated. After 24 days, the AUC of the control group increased slightly by 0.36% to 26,835; the AUC of the solution casting method group decreased by 22.72% to 20,360; and the AUC of the MDO method group decreased by 27.30% to 19,410 (Figure 7a). This shows that intestinal cannula implantation was effective in reducing blood glucose levels due to the effect of duodenal exclusion and gut hormonal changes, and the MDO method group was more effective. Finally, both the solution casting group and the MDO group were found to be effective in reducing weight by approximately 11% after 24 days, and improving the obesity problem (Figure 7b).Using a polytetrafluoroethylene (PTFE)-coated plastic sleeve from EndoBarrier (GI Dynamics) with a duodenal bulb leads to a significant weight loss of approximately 12% and glycemic control, but there are device design complications such as bleeding and perforation [24,43]. As compared with the PTFE-based commercial EndoBarrier, the obtained PVDF-based EndoBarrier device fabrication exhibits significantly reduced blood glucose levels and weight loss of approximately 11% after 24 days of treatment. 

In order to determine how safe the implanted device is, histopathological staining evaluations were performed to highlight any inflammation and tissue damage effects on various tissues such as duodenum, liver and stomach over 4 weeks. As shown in Figure 8, the hematoxylin and eosin (H&E) staining images for duodenum, liver and stomach tissue sections (on Day 28) were did not show significant inflammation or lesions, though the implanted device results in slight chronic inflammation. There is some focal plane erosion that is identical to that in control groups. Compared with normal tissues, after implantation of the intestinal sleeve, the duodenal tissues showed cellular infiltration in the entire intestinal wall layer as well as mild inflammation. As for liver tissue sections, no inflammation and no acute damage or necrosis were observed compared to normal tissue, so this implantation method does not produce liver abscesses. A similar scenario was observed for stomach tissues, i.e., no fibrosis, or acute damage and injury. Overall, the designed PVDF-based EndoBarrier device can be used to create a new safe approach to the piezoelectrical polymer with efficacy in treatment of type 2 diabetes and obesity.

## 4. Conclusions

In this study, PVDF films implanted in the digestive tract of rats for the treatment of type 2 diabetes were successfully prepared by using two different methods: solution casting and MDO. In particular, the PVDF film prepared by the MDO method exhibited a higher degree of structural compactness and better mechanical properties to reduce the risk of damage to the film in the digestive tract. In addition, due to the polarization of PVDF in the process, the phenomenon of crystalline state change was also analyzed by various instruments. After the PVDF was implanted into type 2 diabetic rats, blood glucose levels and body weight were tracked for 24 days and it was found that both had a decreasing effect on controlling blood glucose and weight loss. Overall, this device has demonstrated its potential for use in the control of type 2 diabetes. However, systematic detailed experimental evidence from biopsies of the upper gastro-intestinal tract for implantation of the PVDF-based EndoBarrier can provide insight into the mechanisms behind the increase in glucose metabolism and weight loss, work that is currently in progress. 

## Figures and Tables

**Figure 1 polymers-14-02178-f001:**
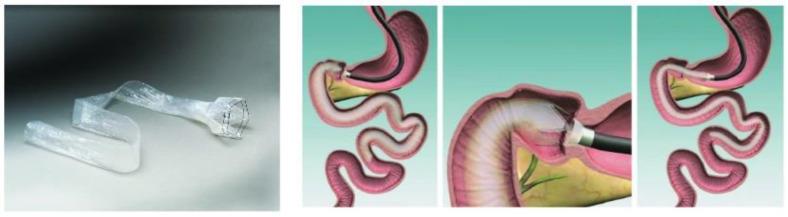
The EndoBarrier device and duodenal implantation (Ruban et al. 2018).

**Figure 2 polymers-14-02178-f002:**
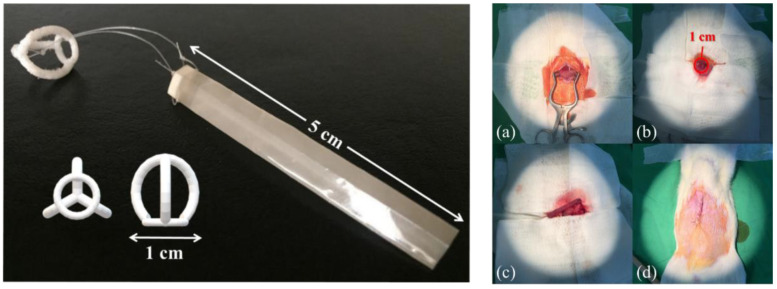
(**Left side**) PVDF film intestinal sleeve implant and, (**Right side** (**a**–**d**)) surgical procedure of film intestinal sleeve implantation in SD rats.

**Figure 3 polymers-14-02178-f003:**
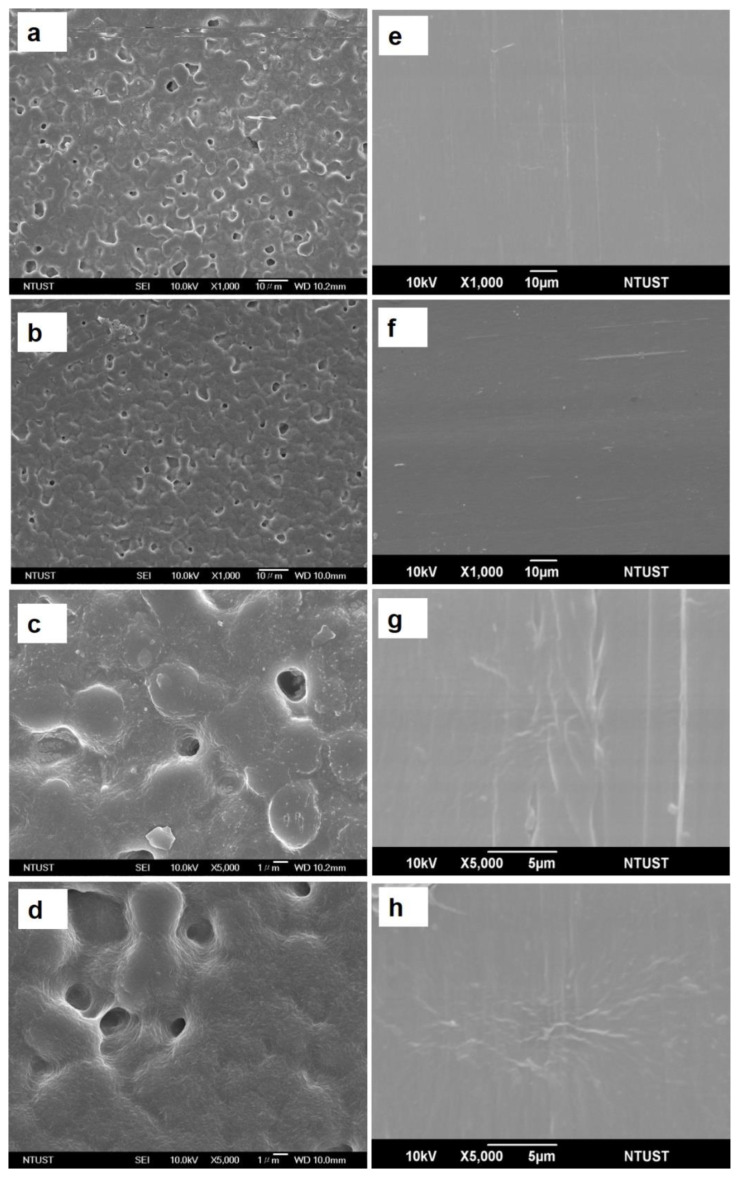
SEM images of a PVDF film formed by solution casting at (**a**) 90 °C and 1000×, (**b**) 100 °C and 1000×, (**c**) 90 °C and 5000×, and (**d**) 100 °C and 5000×; and by MDO at (**e**) 90 °C and 1000×, (**f**) 100 °C and 1000×, (**g**) 90 °C and 5000×, and (**h**) 100 °C and 5000×.

**Figure 4 polymers-14-02178-f004:**
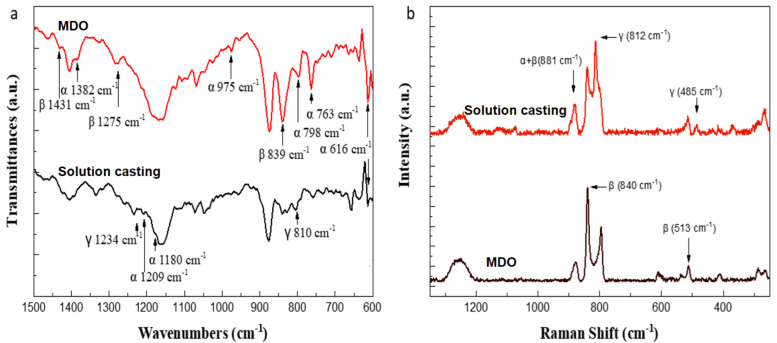
(**a**) FTIR spectra of PVDF films formed by different process methods. (**b**) Raman spectra of PVDF films formed by different process methods.

**Figure 5 polymers-14-02178-f005:**
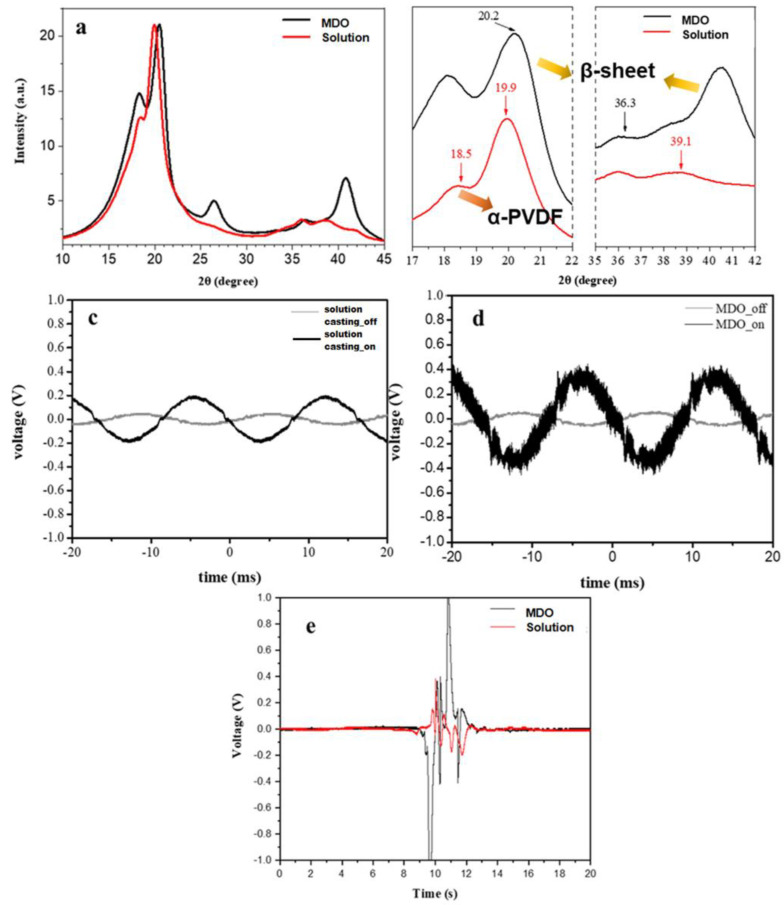
XRD results of PVDF films formed by different process methods: (**a**) overview, (**b**) partial enlargement, piezoelectrical performance with respect to time (**c**,**d**), and with an applied pressure of 1 N/cm^−1^ (**e**).

**Figure 6 polymers-14-02178-f006:**
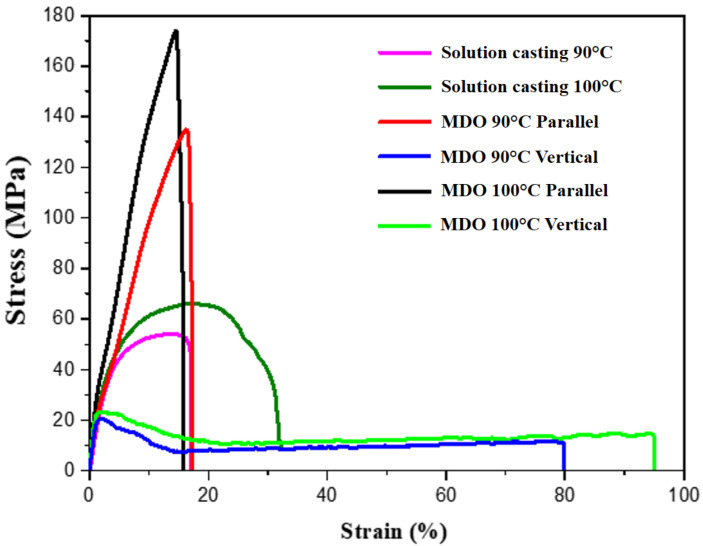
Tensile test results of PVDF films with different process methods.

**Figure 7 polymers-14-02178-f007:**
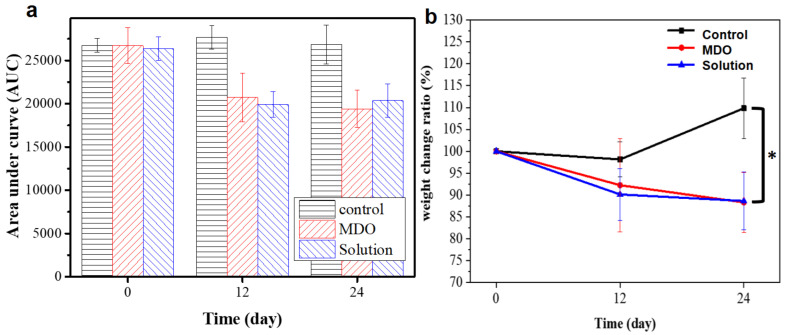
Blood glucose levels (**a**), area under the curve (AUC), and (**b**) weight change rate recorded on day 0, day 12, and day 24. Every group has at least three replications and all data are presented as the mean ± standard error of the mean (SEM). * *p* < 0.05 based on analysis by the unpaired Student’s *t*-test.

**Figure 8 polymers-14-02178-f008:**
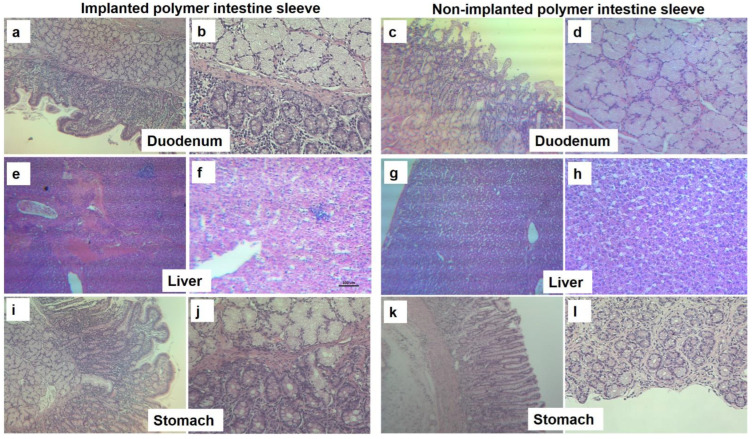
The H&E-stained tissue sections for various implanted tissues. Polymer intestinal sleeve implanted in the duodenum (**a**,**b**) and control duodenum (**c**,**d**); polymer intestinal sleeve implanted in liver tissue (**e**,**f**) and control liver tissue (**g**,**h**); polymer intestinal sleeve implanted in stomach tissue (**i**,**j**) and control stomach tissue (**k**,**l**). Magnification, ×40 in (**a**,**c**,**e**,**g**,**i**,**k**) and ×100 in (**b**,**d**,**f**,**h**,**j**,**l**).

**Table 1 polymers-14-02178-t001:** Peak intensity ratio of each crystalline phase after the peak splitting process.

	α 795 cm^−1^(a.u.)	β 840 cm^−1^ (a.u.)	γ 812 cm^−1^ (a.u.)	β/(α + γ) (%)
Solution casting 90 °C	81.0	142.0	177.0	35.5
Solution casting 100 °C	78.1	126.2	163.3	34.3
MDO 90 °C	154.6	329.6	107.7	55.7
MDO 100 °C	264.0	519.2	110.6	58.1

**Table 2 polymers-14-02178-t002:** Tensile test results of PVDF films with different process methods.

Sample	Yield Point (MPa)	Average Elongation (%)	Young’s Modulus (MPa)
Solution Casting 90 °C	54.14	17.01	22.84
Solution Casting 100 °C	66.08	32.17	38.81
MDO 90 °C Parallel	134.58	16.33	34.67
MDO 90 °C Vertical	20.29	79.77	23.48
MDO 100 °C Parallel	173.72	14.59	36.28
MDO 100 °C Vertical	22.78	94.89	26.94

## Data Availability

Not applicable.

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
