# Peer review of "Poly(vinylidene fluoride) Intestinal Sleeve Implants for the Treatment of Obesity and Type 2 Diabetes"

_polymers, 2022, doi:10.3390/polym14112178_

Round 1

Reviewer 1 Report

The current work is interesting but requires a further modification, as explained below.

Introduction: please add figures regarding the mentioned commercial implementations for a general reader.

L 83 regarding relevance piezoelectric. Please add Prog. Mater. Sci. 2020, 114, 100617 in a general context piezoresistive.

L 104 to remove .. the cure; please rephrase.

For a general, please explain how wire and rolling leads to a film (so 1 to 2D aspect)

L 165 please check “1” needs to be “3”. Conclusion needs to be “4”.

General comment: please add in the experimental section all thus complete details regarding the equipment and settings.

L 247 Please compare to typical tensile curves reported in the literature.

“2036” please check typo.

L 266: sentence in pdf is underlined in red.

Reviewer 2 Report

The manuscript entitled: “Poly(vinylidene fluoride) internal sleeve implant for the treatment of the obesity and type-2 diabetes” submitted to Polymers-MDPI as Article describes two methods of preparation of a PVDF sleave starting from casted solution of by machine direction orientation. Newly prepared films were thermally treated at 90 °C or 100 °C. Afterwards, the materials were characterized by the means of FTIR, RAMAN, SAXS, SEM, mechanical testing and ESI. It was described by authors that the thermal treatment induce change in crystalline structure of polymer giving mainly beta-crystalline structure. At the end animal experiments were performed on SD rats, giving very similar to commercially available commercial Endo-Barrier, a 11% of weight reduction after 24 days.

The topic presented in the manuscript discuss important issue of the civilization diseases and solution to treat them. Search for new and safer approaches to treat the obesity and type-2 diabetes is a important goal for scientific community. However due to some mayor issues, I do not recommend this article for publishing in its current form. 

Here is the list of issues that requires authors attention: 

  1. Introduction: I missed a final paragraph that could state the scope of this work. It would give information to reader about what can be expected in following sections.
  2. Results and discussion, 3.1 characterization of PVDF: Did authors performed cross-section of obtained membranes? This could provide more evident proof of the morphology of the film.
  3. Results and discussion, 3.1 characterization of PVDF, lines187-189: How authors come to a conclusion about the change of crystalline form based only on SEM imaging. Authors should provide a hypothesis that can be later proven, not a direct statement.
  4. Results and discussion: Authors when discussing any results should mention the figure under discussion and later provide their explanation to the results, not as it is now.
  5. Results and discussion, 3.1 characterization of PVDF, lines 211-214: How authors are certain that the Raman spectra are more quantitative then FTIR? Please provide any prove of such statement.
  6. Results and discussion, 3.1 characterization of PVDF, lines 228-232: it was not clear what was the purpose of performing the experiment to confirm piezoelectric performance. Authors should provide some more details about the experimental setup and the reason for performing such experiment.
  7. Results and discussion, 3.1 characterization of PVDF, lines239-239: Authors mentioned values of Young’s modulus without describing form where the data were obtained. I would suggest to present additionally the mechanical testing data in a form of a table, so the results can be easily compared or discussed.
  8. Fig 7: authors should provide scale for all images. Moreover, in order to reduce possible confusion for the reader I would suggest to name the reference tissue images similarly in the text and the figure.
  9. References: I would suggest citing more recent work regarding i.e. characterization of PVDF membrane.

Reviewer 3 Report

The paper shows fabrication of poly(vinylidene fluoride) (PVDF) films an their application as the intestinal sleeve implants. Two solution casting and machine direction orientation (MDO) were used for fabrication of PVDF layers. The prepared samples were examined with FT-IR, Raman spectroscopy, XRD, and piezoelectricity tests. The PVDF films were implanted into the duodenum of type 2 diabetic rats for the treatment of obesity and blood glucose control. The decrease of body weight and blood glucose level in alloxan-induced diabetic rats were observed. The results, presented in the paper, are novel and seem to be attractive for researches involved in development of new biomedical devices. However, some issues have to be improved before the manuscript can be considered for publication. Authors are requested to take into consideration following suggestions:

  1. The recent achievements in the piezoelectric nanogenerators for biomechanical applications should be mentioned in the “Introduction” section (Sustain. Energy Fuels 5 (2021) 6049).
  2. What was the thickness of the PVDF films? What was the thickness distribution along the PVDF film?
  3. The top views of the PVDF films were presented in SEM images (Figure 2). Can be shown the cross sections of the samples additionally?
  4. Authors are requested to explain what are the meaning of the labels “coating_off”, “coating_on”, “MDO_off”, and “MDO_on” given in Figures 4c,d. Why the black and grey curves are different?
  5. What were the impact used to generate the voltage waveforms presented in Figures 4c,d and Figure 4e? What were the values of the excitation frequencies?
  6. The uncertainty of the measurement data, presented in Table 2, is missing.
  7. Tables 1 and 2 should be converted from the images to the editable formats.
  8. What is the exact mechanism behind the decrease of blood glucose after implantation of the intestinal sleeve? Authors are requested to discuss it in detail.
  9. What is the material stability after its implantation? Was this issue studied?

Round 2

Reviewer 1 Report

The authors have update the manuscript but are not aware of the following:

  1. Extra figure in introduction (as shown in the reply letter). You can easily ask for permission to include such figure. There are standard procedures for that.
  2. It is said that no further details regarding analysis can be given because of the patent. The current information is not enough for a scientific publication. Giving information on how you performed e.g. mechanical testing is general knowledge and not patent-critical. Please update.

Reviewer 2 Report

The revised manuscript entitled: “Poly(vinylidene fluoride) internal sleeve implant for the treatment of the obesity and type-2 diabetes” submitted to Polymers-MDPI as Article for second revision.

After careful analysis of the response of authors to the remarks I found the explanation suitable for most of issues. In my opinion this article can be published after adding a separate paragraph regarding the scope of the work at the end of the introduction, i.e.:

The aim of this work is focused on a PVDF sleave prepared by two methods: solution casted or machine direction orientation. Newly prepared films were thermally treated and afterwards, characterized by the means of FTIR, RAMAN, SAXS, SEM, mechanical testing and ESI. The thermal treatment induced change in crystalline structure of polymer giving mainly beta-crystalline structure due to transformation of alpha-crystalline structure. At the end animal experiments were performed on SD rats, giving very similar to commercially available commercial Endo-Barrier, a 11% of weight reduction after 24 days”.

Reviewer 3 Report

The manuscript has been revised and now can be accepted for publication.

Author Response

Thank you very much!

Round 3

Reviewer 1 Report

The paper is ready for publication. A general comment for non-open access contributions. It is not the authors who share the rights but the publisher.